# The Impact of Epithelial Inflammation in Membrane Remnants on the Outcome of Tympanoplasty

**DOI:** 10.3390/medsci13020073

**Published:** 2025-06-07

**Authors:** María Fernanda Galindo-Tapia, Alejandro Esteban Deras-Quiñones, Itzel Maria Montoya-Fuentes, Eduardo Osiris Madrigal-Santillán, Ángel Morales-González, Naria A. Flores-Fuentes, Liliana Anguiano-Robledo, Raúl Rojas-Martínez, Beatriz Montaño-Velázquez, José A. Morales-González

**Affiliations:** 1Instituto Mexicano del Seguro Social, Centro Médico Nacional la Raza, Hospital General “Dr. Gaudencio González Garza”, Servicio de Otorrinolaringología, Mexico City 02990, Mexico; galindotapiamafer@gmail.com; 2Instituto Mexicano del Seguro Social, Centro Médico Nacional La Raza, Hospital de Especialidades “Dr. Antonio Fraga Mouret”, Servicio de Patología, Mexico City 02990, Mexico; dr.alexdequi@gmail.com (A.E.D.-Q.); anatopatoitzelmmf@gmail.com (I.M.M.-F.); 3Laboratorio de Medicina de Conservación, Escuela Superior de Medicina, Instituto Politécnico Nacional, Plan de San Luis y Díaz Mirón, Colonia Casco de Santo Tomás, Alcaldía Miguel Hidalgo, Mexico City 11340, Mexico; eomsmx@yahoo.com.mx (E.O.M.-S.); romr65@yahoo.com.mx (R.R.-M.); 4Escuela Superior de Cómputo, Instituto Politécnico Nacional, Unidad Profesional “A. López Mateos”, Mexico City 07738, Mexico; anmorales@ipn.mx (Á.M.-G.); nafloresf@ipn.mx (N.A.F.-F.); 5Laboratorio de Farmacología Molecular, Escuela Superior de Medicina, Instituto Politécnico Nacional, Plan de San Luis y Díaz Mirón, Col. Casco de Santo Tomás, Del. Miguel Hidalgo, Mexico City 11340, Mexico; languianorobledo@yahoo.com.mx

**Keywords:** chronic otitis media, tympanic perforation, tympanoplasty, epithelial inflammation

## Abstract

Background: Chronic otitis media (COM) with tympanic perforation sometimes requires tympanoplasty. Many factors can interfere with surgical success; however, the histological status of the remaining epithelium of the perforation has not been studied as a risk factor for surgical failure. Methods: This was an observational, longitudinal, and analytical study in patients with COM, candidates for tympanoplasty who met the inclusion criteria, between August and December 2024. Tympanoplasty was performed, and the tympanic ring epithelium was sent for histological analysis. After 30 days, closure or non-closure of the perforation was determined, and the results were collected. Descriptive and analytical statistics were performed according to data distribution using the SPSS 26.0 statistical package. Results: Twenty subjects were included, 80% with tubal dysfunction, 60% with central perforation, and 65% with medium-sized. In total, 13 were successful, and 7 failed. Histopathological analysis revealed dystrophic calcification, chronic lymphocytic infiltrate, histiocytic infiltrate, fibrosis, loose keratin sheets, metaplasia, and spongiosis. The logistic regression model showed an OR of 7.3 for marginal perforation and 3.4 for the OPSS score. Of the patients with surgical failure, 57.4% had epithelial inflammation. Conclusions: epithelial inflammation affected surgical success in more than 50%.

## 1. Introduction

Chronic otitis media (COM) is a very common condition in our environment. It is defined as an inflammatory process of the middle ear mucosa that lasts more than 3 months [1], with purulent discharge and perforation of the tympanic membrane [2,3]. Tympanic perforation is defined as a rupture of the tympanic membrane, creating a connection between the external auditory canal and the middle ear [4], making the ear more susceptible to infection and hearing loss. The prevalence of COM in the world’s population ranges from 1% to 6%, up to 11% in developing countries such as Mexico and 57% in countries with extreme poverty [5,6,7]. There are more than 150 million people with tympanic perforation worldwide [5], with a global incidence of 4.8 cases per 1000 people per year. Tympanic perforation varies between 5% and 30% [4]. In Mexico, it is estimated that 0.5% of patients over 15 years of age have chronic otitis media, and 4% have tympanic perforation [1]. In the Otolaryngology and Head and Neck Surgery Department of the General Hospital of the National Medical Center “La Raza”, 82% of consultations are for COM.

The main cause of perforation is middle ear infection, followed by trauma (87.5% vs. 12.5%) [8]. Recurrent infections, bacterial colonization, aural ventilation dysfunction, tubal dysfunction, or immunodeficiency can affect closure [9], as can myringotomy, ventilation tubes, and hyperbaric oxygen therapy [10]. In rarer cases, it can be due to optic herpes zoster, tuberculosis, and aggressive otitis externa [11].

The tympanic membrane separates the middle ear from the external ear. It is oval in shape, with a vertical diameter of 9–10 mm and a horizontal diameter of 8–9 mm. Macroscopically, it is divided into two parts: the pars flaccidum superior to the malleus and the pars tensa inferiorly. It is composed of three layers [12]: an outer layer composed of simple stratified squamous epithelium, which is continuous with the skin of the ear canal; a middle fibrous layer called the lamina propria, formed by radial and circular fibers; and finally, the inner layer composed of pseudostratified low cuboidal epithelium, which is continuous with the mucosa of the middle ear [13].

When the tympanic membrane is perforated, it tends to close spontaneously. Unlike the skin, it closes from the outside in, due to the proliferation of stratified squamous epithelium in the lateral portion, which migrates from the remnant to the central portion of the perforation [12,13,14]. Once the outer layer closes the defect, the fibrous layer and mucosa regenerate beneath it [15]. This regeneration can be affected by several factors, including chronic inflammation, epithelialization of the perforation margins, insufficient blood supply, or growth factor deficiency [16]. If a tympanic membrane perforation does not heal on its own within 12 weeks, it is unlikely to heal and should be treated with tympanoplasty [7,8,9], which is the reconstruction of the tympanic membrane to prevent disease progression and, to a lesser extent, to restore hearing. Most studies on surgical success report success rates between 80% [17] and 90% in ears with normal ventilation [9]; however, in cases with chronic inflammation of the middle ear, the outcome tends to be less successful. Moyakan et al. mention that 20% of unsuccessful tympanoplasties are associated with incomplete integration, graft retraction, or graft displacement [17].

Throughout history, studies have been conducted on the histological changes in the ear mucosa resulting from chronic otitis media. In 1970, the presence of mucosal hyperplasia and mucus hypersecretion was demonstrated [18]. In 2013, mucosal hyperplasia, edema, polypoid changes, tympanosclerosis, and mucosal granulomas were identified [7]. However, there are few histological studies of the epithelium of the perforation remnant. In 2022, a study was conducted on the epithelial remnant of the perforation and the surrounding middle ear mucosa. The study found normal epithelium in 10.26% of cases, hypertrophied epithelium with chronic inflammatory infiltrate in 48.72%, tympanosclerosis with chronic inflammatory infiltrate in 20.51%, edematous hypertrophic mucosa in 17.95%, and keratosis with vascular stroma in 2.56%, with a surgical success rate of 84.62% at 3 months [6]. However, data on inflammation were not cross-referenced with surgical success. This is the reason for the interest in conducting this study, to determine whether there was a relationship between the presence of epithelial inflammation and the success of surgical closure.

## 2. Materials and Methods

This was an observational, longitudinal, analytical study with non-probability convenience sampling. It was conducted in the otolaryngology department of the General Hospital “Dr. Gaudencio González Garza” at the National Medical Center “La Raza “ of the Mexican Social Security Institute.

Patients were selected via the following inclusion criteria: a diagnosis of chronic otitis media with tympanic membrane perforation, no prior surgical treatment, age over 18 years, preoperative CT and audiometric studies, and no requirement for mastoidectomy or any other procedure. We classified the clinical severity of disease according to the OOPS index (Ossiculo-plasty Outcome Parameter Staging) and MERI (Middle Ear Risk Index). Once selected, the informed consent form was read, reviewed, and signed by each individual. Between August 2024 and December 2024, the tympanoplasties were performed, all endoscopic transcanal, using tragus cartilage grafts, by the same surgeon.

During the procedure, lidocaine with epinephrine was infiltrated into the tragus and the posterior wall of the external auditory canal. Subsequently, an incision was made in the tragus with a 15 number scalpel and dissected to harvest the cartilage graft, and it was closed with 4–0 nylon sutures. Under 0-degree endoscopic vision, with a 45-degree angle, an approximately 1 mm incision was made in the epithelium surrounding the perforation, shaped like a postage stamp. The incisions were then joined, and the annulus was detached and removed with alligator forceps; then, using a Rosen scalpel, the surrounding epithelium was removed 2 mm from the outer layer by gently scraping the membrane. The tissue was removed with alligator forceps. Once the epithelium was removed from the remnant, hydrocortisone-impregnated Gelfoam was placed in the middle ear cavity. Then, the cartilage graft was placed at the perforation site, and finally more Gelfoam was placed over the graft.

Once the tissue was collected, it was placed in a sealed container immersed in 10% formaldehyde for fixation for at least 24 h while it was sent to the pathology department. Once there, each sample was passed through an alcohol train for dehydration, embedded, and cut into paraffin blocks approximately 3 μm thick. The slides were mounted and stained with hematoxylin and eosin, then covered with coverslips for microscopic analysis using 4×, 10×, and 40× objectives to evaluate the presence of inflammatory cells per field. Sample analysis was performed by two pathologists in a blinded manner, with a pre-standardized approach for reporting histopathological changes.

Patients were seen again at 10, 30, and 90 days. At the first appointment, the sutures were removed from the graft harvesting site, and the success or failure of the tympanoplasty was determined at the second appointment with no changes at 30 and 90 days. Histopathology results were also collected to determine the presence or absence of epithelial inflammation in the perforation remnant.

Once the data were collected, statistical analysis was performed. Mean and standard deviation were used for normally distributed quantitative variables; median and interquartile ranges were used for freely distributed variables. Finally, qualitative variables were presented as frequencies and percentages. Bivariate analysis was performed using the chi-square test to compare the presence or absence of epithelial inflammation versus tympanoplasty success or failure. The different characteristics between the two study groups were compared using the Student *t* test for normally distributed quantitative variables; the Mann–Whitney U test was used for freely distributed variables; and the chi-square test was used for dichotomous qualitative variables. Variables that showed statistical significance were included in a linear regression model to determine risk factors for epithelial inflammation and those that directly impacted surgical success. This model was performed using the SPSS 26.0 statistical package, and *p* values < 0.05 were considered significant.

This study was evaluated and approved by the local Research and Ethics Committee. This study adhered to the General Health Law and its Regulations on Health Research, as well as the Standards of the Mexican Social Security Institute. It complied with the provisions of the National Institute for Transparency, Access to Information, and Protection of Personal Data, as well as NOM 012-SSA3-2012. This study also adhered to the current ethical considerations contained in the Nuremberg Code, the Council for International Organizations of Medical Sciences (CIOMS), the Belmont Report, the Declaration of Helsinki, and the international guidelines for medical research involving human subjects adopted by the WHO and the Council for International Organizations for Research Involving Human Subjects. There are no conflicts of interest for conducting this study or its publication.

## 3. Results

Twenty patients were included, 15 women (75%) and 5 men (25%). The mean age was 56 years with a SD of 10.9 years, with a minimum age of 37 and a maximum of 78. The mean for the OOPS score was 2.45 (SD 1.5), and for the MERI score, it was 4 (1.8). The total presurgical hearing average presented a mean of 45.5 dB (SD 15) with a minimum of 23 dB and a maximum of 86 dB. Regarding tubal dysfunction, 16 patients presented it (80%); 50% of the affected ears were right and 50% left, and 12 (60%) presented central perforation. About size, 3 (15%) had a small perforation (<30%), 13 (65%) had a medium perforation (30–60%), and 4 (20%) had a subtotal perforation (>60%). Half of the subjects (n = 10) had well-pneumatized mastoids and the rest (n = 10) had sclerotic mastoids (Table 1).

Regarding postoperative care, 18 subjects (90%) reported adequate postoperative dry ear care, while 2 of them experienced water ingress. Thirteen subjects (65%) were successful, and seven (35%) had surgical failure within the first 30 days. Histopathological analysis of the samples reported six cases (30%) with dystrophic calcification, three (15%) with leukocytic infiltrate, and three (15%) with histiocytic infiltrate; one patient presented metaplasia of squamous epithelium with cuboidal epithelium, and one additional patient presented spongiosis. Figure 1 shows the rest of the results with fibrosis, loose keratin sheets, and normal epithelium. Cases with leukocyte infiltrate, spongiosis, and the presence of histiocytes were classified as having inflammation; the rest were considered without epithelial inflammation (Figure 1).

Table 2 summarizes the baseline characteristics of the patients according to the presence or absence of epithelial inflammation of the remnant of the tympanic perforation, where no statistically significant differences were observed between the groups.

Cross-tabulating the presence of epithelial inflammation of the perforation remnant versus surgical outcome showed that 50% of patients with epithelial inflammation had surgical failure, while only 25% of those without inflammation of the remnant had failure. Of the total patients with surgical failure, 57.4% had epithelial inflammation.

Regarding the various histological changes reported, the most successful was dystrophic calcification, with 31% of the total successful cases categorized as absence of epithelial inflammation, while the most unsuccessful was histiocytosis, with 29% of the failed cases categorized as epithelial inflammation (Figure 2 and Figure 3).

Fisher’s exact test to calculate the relationship between the inflammatory status of the epithelial remnant and surgical outcome yielded a *p* value of 0.356, with no statistical significance. However, when calculating the OR, the presence of inflammation in the remnant yielded a result of 1.5 (95% CI 0.697–3.22).

The multivariate logistic regression model to determine predictors of surgical failure showed the presence of epithelial inflammation to be the primary variable, with an OR of 3.46 (95% CI 0.37–31.94), followed by median perforation with an OR of 2.6 (95% CI 0.158–43.75), and sclerotic mastoid with an OR of 2.4 (95% CI 0.158–39.09) (Table 3).

A logistic regression model was performed to determine the predictive factors for the presence of inflammation of the epithelial remnant of the tympanic perforation, finding that the variables that presented a higher probability of risk for presenting inflammation were marginal perforation with an OR of 7.3 (95% CI 0.145–371.58) and a high OOPS index with an OR of 3.4 (95% CI 0.65–18.712) (Table 4).

## 4. Discussion

The primary objective of this study was to determine whether there was a relationship between epithelial inflammation of the perforation remnant and surgical success. We also evaluated the success rate and factors associated with surgical failure.

We estimated the effect size and detected a power of 77% (group with success, 25% vs. group without success, 57.7%), and it tells us that relationships between variables or the differences between groups were high [19].

In our population, the closure rate was low, as described in other studies; it has been considered that the severity of the disease in these types of patients can be the main factor contributing to a low closure rate [20]. The mean for the OOPS score was 2.45 (SD 1.5), and for the MERI score, it was 4 (1.8). In the group of patients with surgical failure, the mean for OOPS was 3.7 (SD 0.95), and for MERI it was 5.1 (SD 1.3); that means that most of the patients presented moderate disease. The disease was classified into three stages: mild, moderate, and severe. The OOPS score ranged from 1 to 3, 4 to 6, and 7 to 9, and the MERI score ranged from 1 to 3, 4 to 6, and 7 to 12, which means that most of the patients presented moderate and severe disease [21,22].

Regarding surgical success, our overall success rate was 65%, similar to that reported by Fukuchi [3]; however, it was lower than that reported in most studies. Asfaha et al. conducted a study with 92 patients, obtaining an overall success rate of 89.1%. However, in tympanoplasties performed with cartilage grafts, they reported a failure rate of 31.3% [23]. This failure rate is higher than ours, which was 25%, considering that all tympanoplasties were performed with cartilage grafts. We selected cartilage graft, taking into account the fact that in our sample the majority of patients presented tubal dysfunction (80%) and some authors indicate that in these cases is better to use cartilage graft [24,25], because it can be more resistant and not retract post-surgery as in the case of fascia [26]. Chen Chin Kuo evaluated the success of endoscopic tympanoplasty with cartilage, a technique similar to ours, reporting a success rate of 86.3% [27], higher than ours. Age and sex were excluded from the analysis because they were not associated or statistically significant with the surgical outcome or with the inflammatory status of the tympanic membrane. Similarly, Momin et al., in their study of risk factors for the success of tympanoplasty, reported that age was not associated with closure of the perforation (OR 1.04, *p* = 0.31) [28]. Our study presented seven patients with surgical failure, 85% of whom presented tubal dysfunction. Moneir et al. studied the correlation between tubal dysfunction and the results of type 1 tympanoplasty, finding that 60% of patients with tubal dysfunction presented graft retraction and residual perforations, while 20% persisted with the same perforation [29]. Among other risk factors, the presence of postoperative infection was found; the two patients who presented this condition suffered surgical failure, while Asfaha reported that of the patients who presented postoperative otorrhea, 28.6% had surgical failure [23]. In their logistic regression model, they demonstrated that the most important predictor of failure was the presence of postoperative infection with an odds ratio of 9.6 (95% CI 1.1–88.9) [23], meaning that patients with postoperative otorrhea had an 8.6-fold higher risk of surgical failure compared to those without otorrhea. Fukuchi mentioned that the site and size of the perforation did not present statistical significance as a risk factor for the success of tympanoplasty [3], as did Abdelhameed, who studied whether the size of the perforation and graft had an impact on the surgical result by dividing the sample according to the size of the perforation into three groups and then correlating them obtaining non-significant *p* values [30]. In contrast to these results, when performing the multivariate logistic regression model, we found that medium-sized tympanic perforation (30% to 60%) did represent a risk factor for surgical success with an OR of 2.62 (95% CI 0.158–43.7). Regarding epithelial inflammation of the perforation remnant, it was found that of the seven patients who presented surgical failure at 30 days, 57.4% presented epithelial inflammation, which is close to the hypothesis proposed in the present study, in which it was postulated that the presence of inflammation of the epithelium of the tympanic perforation would influence the success of tympanoplasty in at least 60% of patients with chronic otitis media. On the other hand, Mili et al. reported chronic inflammatory infiltrate in 69.23%, with a surgical success rate of 84.6% [6]. Tissue inflammation is an essential biological response of the body to injury, infection, or cellular damage. Its role in tissue regeneration is that it facilitates repair processes; however, if chronic, it can interfere with proper regeneration. During acute inflammation, immune cells such as neutrophils, macrophages, and monocytes migrate to the site of injury as a response to mediators released, such as cytokines, acute-phase proteins, and chemokines [31], where they eliminate pathogens and cellular debris, in addition to secreting cytokines and growth factors (such as TGF-β and VEGF) that promote cell proliferation and angiogenesis [32]. However, prolonged or chronic inflammation can lead to fibrosis and alterations in tissue architecture. Wynn and Vannella reported that sustained activation of macrophages, along with the production of proinflammatory cytokines (TNF-α, IL-1β), can inhibit cell differentiation and promote excessive deposition of the extracellular matrix [33]. Taking into account these observations and the results obtained in this study, it is speculated that the presence of chronic inflammation and changes in architecture are closely related to the failure of perforation closure.

Apart from this study, no similar studies were found that evaluated the impact of epithelial inflammation on the surgical outcome of tympanoplasty. However, Leffers, in his article on the immunomodulatory response of epithelial cells in otitis media, evaluated the immunohistochemical changes that occur in the epithelium after inoculation with *Haemophilus influenzae*, finding an increase in IL-1β, IL-6, IL-8, and TNFα [34]. This could open a new line of research in which immunohistochemical studies of the epithelial remnant of the perforation can be performed.

The multivariate logistic regression model showed that the factor that most influenced the success of tympanoplasty was the presence of epithelial inflammation of the perforation remnant, with an odds ratio of 3.4 (95% CI 0.37–31.9), indicating that the presence of epithelial inflammation increases the likelihood of surgical failure by 2.4 times.

In turn, the regression model for predictors of epithelial inflammation indicated that the factors with the greatest risk were the OOPS score, which indicates the state of the mucosa, and the presence of marginal perforation.

Regarding the state of the mucosa, we standardized it with the OOPS score, in which a lower score indicated better condition of the middle ear mucosa. Our study showed low scores, that is, ears with little mucosal damage derived from the previous selection of patients; however, when analyzing its association with the presence of inflammation of the remnant of the perforation, it showed an odds ratio of 3.4 (95% CI 0.65–18.7), which means that for each unit increased in the OOPS index, the probability of having inflammation of the epithelial remnant increases 2.4 times. On his part, Han Yu compared the impact of different states of the mucosa with the result of tympanoplasty; he did so by classifying patients into four groups, group 1, healthy mucosa; group 2, mucosa with little edema; group 3, with moderate edema; and group 4, with severe edema, finding that most cases of residual perforation occurred in groups 3 and 4 with a percentage of 91.1% and 84.6% [35], respectively.

Limitations of this study: We understand that the sample size is small and that we took many variables into account. However, the statistical power is 77%, and since this is the first study to address epithelial inflammation of the epithelial remnant of the perforation and because significant results were found in the sample size used, we believe this is the beginning of a path to follow.

## 5. Conclusions

Epithelial inflammation of the perforation remnant affected surgical success in more than 50% of patients. However, the wide confidence intervals suggest that the sample size was insufficient, so further research could be carried out and the sample expanded to obtain more statistically reliable results.

Among the factors that favor the presence of epithelial inflammation of the perforation remnant are the presence of marginal perforation, a perforation size greater than 30%, and a high OOPS index. This leads us to conclude that the presence of epithelial inflammation of the perforation remnant negatively impacts surgical outcome. However, it is not the only factor involved in tympanoplasty failure. It would be worth extending this study and taking into account factors such as disease progression time and the presence of comorbidities that were not considered.

## Figures and Tables

**Figure 1 medsci-13-00073-f001:**
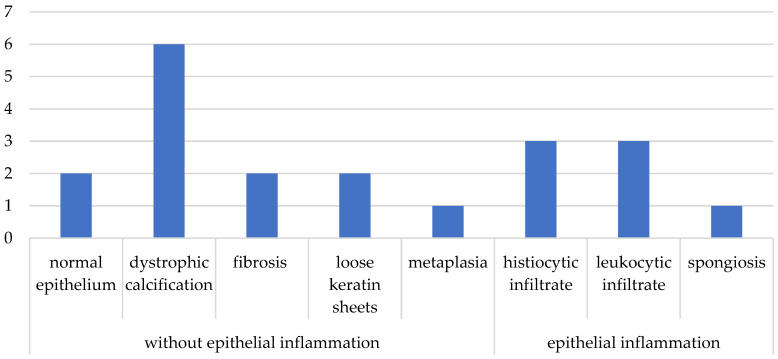
Histopathological results.

**Figure 2 medsci-13-00073-f002:**
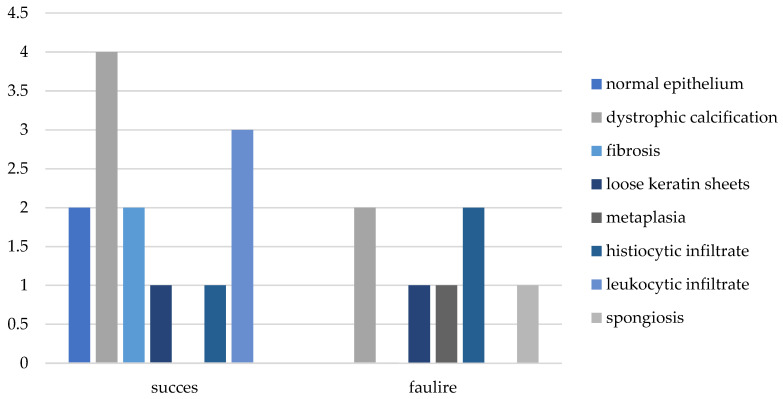
Histopathological results vs. surgical outcome.

**Figure 3 medsci-13-00073-f003:**
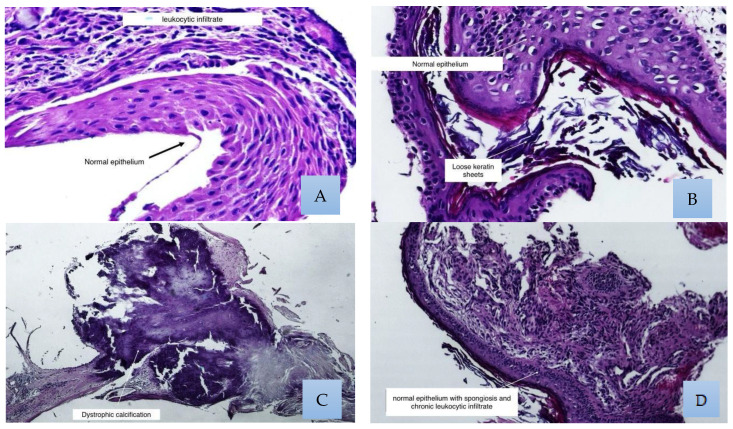
Histological findings were generally found as a collage in the same field; these photos show the most representative examples. (**A**) Normal stratified squamous epithelium with chronic leukocytic infiltrate. (**B**) In this photograph, we can see loose keratin sheets. (**C**) One of the most frequent findings was dystrophic calcification. (**D**) Normal epithelium with spongiosis.

**Table 1 medsci-13-00073-t001:** Baseline characteristics of patients with tympanic perforation.

Variable	n = 20
Sex, women, N (%)	15 (75)
Age, years, mean, (SD)	56.7 (10.9)
ATA, dB, mean (SD)	45.5 (15)
OOPS index, mean (SD)	2.45 (1.5)
Tubal dysfunction, present, N (%)	16 (80)
Ear, right, N (%)	10 (50)
Perforation site, central, N (%)	12 (60)
Perforation size	
<30% N (%)	3 (15)
30–60% N (%)	13 (65)
>60% N(%)	4 (20)
Mastoids, well pneumatized, N (%)	10 (50)

N: number of patients. SD: standard deviation. ATA: auditory tonal average. dB: decibel.

**Table 2 medsci-13-00073-t002:** Baseline characteristics of patients with epithelial inflammation vs. patients without epithelial inflammation.

Variable	Epithelial Inflammation		
	Present	Absent	
	n = 12	n = 8	*p*
Sex, women, N (%)	10 (83.33)	5 (62.5)	0.296 `
Age, years, mean, (SD)	52 (13.3)	52.6 (14.6)	0.427 **
ATA, dB, mean (SD)	50.8 (15.8)	42.4 (12.9)	0.313 **
OOPS index, mean (SD)	3.13 (1.12)	2(1.7)	0.145 **
Tubal dysfunction, present, N (%)	10 (83.33)	6 (75)	0.535 `
Ear, right, N (%)	7 (58.33)	3 (37.5)	0.325 `
Perforation site, central, N (%)	7 (58.33)	5 (62.5)	0.612 `
Perforation size			
<30% N (%)	2 (16.66)	1 (12.5)	0.651 °
30–60% N (%)	8 (66.66)	5 (62.5)	
>60% N (%)	2 (16.66)	2 (25)	
Mastoids, well pneumatized, N (%)	5 (41.66)	5 (62.5)	0.325 `

N: number of patients. SD: standard deviation. ATA: auditory tonal average. dB: decibel. ** U Mann–Whitney. ` Fisher test. ° Chi^2^.

**Table 3 medsci-13-00073-t003:** Multivariate model of factors associated with successful tympanoplasty outcome.

	Beta	Standard Error	Odds Ratio	CI 95% Lower	CI 95% Upper
Central perforation	−1979	1564	0.138	0.006	2962
30–60% perforation	0.966	1435	2627	0.158	43,759
Sclerotic mastoid	0.912	1405	2489	0.158	39,092
Presence of inflammation	1244	1133	3469	0.377	31,940

**Table 4 medsci-13-00073-t004:** Multivariate model for factors associated with the presence of epithelial remnant inflammation.

	Beta	Standard Error	Odds Ratio	CI 95% Lower	CI 95% Upper
OOPS index	1249	0.857	3489	0.650	18,712
Tubal dysfunction present	−3353	2638	0.035	0.000	6163
Sclerotic mastoid	−4044	2545	0.018	0.000	2569
Left ear affected	−2130	1558	0.119	0.006	2518
Marginal perforation	1994	2002	7342	0.145	371,586
Perforation 30–60%	−7506	4873	0.001	0.000	7731
Perforation >60%	−3854	2867	0.021	0.000	5850
Inadequate ear care	−4025	3017	0.018	0.000	6606

## Data Availability

If you need the database, please send it to galindotapiamafer@gmail.com.

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
