# Peer review of "The Impact of Epithelial Inflammation in Membrane Remnants on the Outcome of Tympanoplasty"

_medsci, 2025, doi:10.3390/medsci13020073_

Round 1
Reviewer 1 Report
Comments and Suggestions for Authors
Thank you for the opportunity to review this manuscript, titled “Impact of epithelial inflammation of the membrane remnant on the outcome of tympanoplasty”. Please consider the following comments
- Please describe the cause of the tympanic membrane perforation (infection, trauma, placement of a ventilation tube, etc.).
- The description of the surgical procedure is insufficient.
- A postoperative observation period of only 30 days is too short. A longer follow-up period is necessary to assess surgical outcomes more accurately.
- The manuscript states that the procedures were performed by “the same surgical team.” Was the surgery actually performed by a single surgeon? If multiple surgeons were involved, were there differences in their levels of experience?
- The perforation closure rate (65%) is still considered low despite being the first surgery. Are there any technical factors or differences in the surgeons' years of experience that may have contributed to this?
- In this study, cartilage was used as the transplant material. Could you explain why cartilage was selected without using soft tissues such as fascia in the initial surgery?
- This study is very interesting in that it focuses on inflammation of the tympanic membrane epithelium. However, the discussion is limited to describing statistical trends. Inflammatory cells are also thought to be involved in tissue regeneration and repair, and inflammation and regeneration/repair are known to be deeply related. Given the pathological histological evaluation conducted, it would be desirable to include additional discussion from a biological perspective, even if speculative, regarding the biological effects of inflammation on tissue regeneration (e.g., cytokines, cell proliferation factors, immune responses, etc.).
Author Response
Thank you for the opportunity to review this manuscript, titled “Impact of epithelial inflammation of the membrane remnant on the outcome of tympanoplasty”. Please consider the following comments
- Please describe the cause of the tympanic membrane perforation (infection, trauma, placement of a ventilation tube, etc.).
Answer: The main cause of tympanic membrane perforation cases is of infectious origins, as described in Introduction.
- The description of the surgical procedure is insufficient.
Answer: In the previous manuscript, we described only the procedure for removing the remnant. Considering this observation, we describe the procedure in more detail on section material and methods.
“During the procedure, lidocaine with epinephrine were infiltrated into the tragus and the posterior wall of the external auditory canal. Subsequently, an incision was made in the tragus with a 15 number scalpel and dissected to harvest the cartilage graft and closed it with 4-0 nylon sutures. Under 0-degree endoscopic vision, with a 45-degree angle an approximately 1-mm incision was made in the epithelium surrounding the perforation, shaped like a postage stamp. The incisions were then joined, and the annulus was detached and removed with alligator forceps; then, using a Rosen scalpel, the surrounding epithelium was removed 2 mm from the outer layer by gently scraping the membrane. The tissue was removed with alligator forceps. Once the epithelium was removed from the remnant, hydrocortisone-impregnated Gelfoam was placed in the middle ear cavity, then the cartilage graft was placed at the perforation site and finally more Gelfoam was placed over the graft”
- A postoperative observation period of only 30 days is too short. A longer follow-up period is necessary to assess surgical outcomes more accurately.
Answer: In this study, we continue the observation post-operation at 90 days and no changes were found compared to 30 days. Therefore, the results were the same at 30 and 90 days. Clarification was made on page 3.
- The manuscript states that the procedures were performed by “the same surgical team.” Was the surgery actually performed by a single surgeon? If multiple surgeons were involved, were there differences in their levels of experience?
Answer: the surgeries were always performed by the same surgeon.
- The perforation closure rate (65%) is still considered low despite being the first surgery. Are there any technical factors or differences in the surgeons' years of experience that may have contributed to this?
Answer: We consider the closure rate is related with type of graft; some studies made with cartilage grafts reported most failure rate, as we reported on discussion in the page 7.
" In our population, closure rate is low as described in other studies; it has been considered that the severity of the disease in these types of patients can be the main factor to contribute to a low closure rate (Bhadesia et al 2020). The most of our patients had severe disease.
Regarding surgical success, our overall success rate was 65%, similar to that reported by Fukuchi (Fukuchi et al 2006); however, it is lower than that reported in most studies. Asfaha et al. conducted a study with 92 patients, obtaining an overall success rate of 89.1%. However, in tympanoplasties performed with cartilage grafts, they reported a failure rate of 31.3% (Asfaha et al 2025). This failure rate is higher than ours, which was 25%, considering that all tympanoplasties were performed with cartilage grafts. "
- In this study, cartilage was used as the transplant material. Could you explain why cartilage was selected without using soft tissues such as fascia in the initial surgery?
Answer: The use of cartilage graft in primary surgery was considered because a cartilage graft can be more resistant and not retract post-surgery like in the case of fascia, which is thinner and has a higher probability of retraction (Mucha et al 2023). We chose cartilage graft because it was ideal for our type of patients with severe disease. We made this clarification on page 7.
- This study is very interesting in that it focuses on inflammation of the tympanic membrane epithelium. However, the discussion is limited to describing statistical trends. Inflammatory cells are also thought to be involved in tissue regeneration and repair, and inflammation and regeneration/repair are known to be deeply related. Given the pathological histological evaluation conducted, it would be desirable to include additional discussion from a biological perspective, even if speculative, regarding the biological effects of inflammation on tissue regeneration (e.g., cytokines, cell proliferation factors, immune responses, etc.).
Answer: Keeping your observation in mind, we added a paragraph explaining that points in Discussion
“Tissue inflammation is an essential biological response of the body to injury, infection, or cellular damage. Its role in tissue regeneration is that it facilitates repair processes; however, if chronic, it can interfere with proper regeneration. During acute inflammation, immune cells such as neutrophils, macrophages, and monocytes migrate to the site of injury as a response to mediators released, such as cytokines, acute phase proteins, and chemokines (Hannoodee, et al 2024), where they eliminate pathogens and cellular debris, in addition to secreting cytokines and growth factors (such as TGF-β and VEGF) that promote cell proliferation and angiogenesis (Eming et al 2014) . However, prolonged or chronic inflammation can lead to fibrosis and alterations in tissue archi-tecture. Wynn and Vannella reported that sustained activation of macrophages, along with the production of proinflammatory cytokines (TNF-α, IL-1β), can inhibit cell dif-ferentiation and promote excessive deposition of extracellular matrix (Wynn and Vannella 2016) . Taking into account these observations and the results obtained in this study, it is speculated that the presence of chronic inflammation and changes in architecture are closely related to the failure of perforation closure.”
Reviewer 2 Report
Comments and Suggestions for Authors
With all respect to the authors, but I must say that I find the text as it is rather disappointing....No matter how good the introduction is written, no matter how well the results are depicted ( I mean the diagramms), the total number of the participants is very low. The statisticsal analysis itself is not enough to " cure" such a weakness.... Its part of the article is analyzed above:
Introduction: It is o.k. I do not have specific comments.
Materials: I am sorry, but you are trying to make out some conclusions, reporting that you have studied on;ly 20 subjects. The number of the patients suffering from Tube-dysfunction is very (very.....) low
Discussion: Too short for such a theme. It MUST be re-rewritten.
I do not make any further comments on the References.....
Author Response
- With all respect to the authors, but I must say that I find the text as it is rather disappointing....No matter how good the introduction is written, no matter how well the results are depicted ( I mean the diagramms), the total number of the participants is very low. The statisticsal analysis itself is not enough to " cure" such a weakness.... Its part of the article is analyzed above:
Answer: We understand that the sample size is small; however, this is the first study to address epithelial inflammation of the epithelial remnant of the perforation. We believe this the beginning of a path to follow, not to mention that we plan to continue with this study, expand it further, and taking these results as a preliminary.
We add a paragraph for limitations of the study on discussion , page 9.
- Introduction: It is o.k. I do not have specific comments.
Answer: Thank you for the observation.
- Materials: I am sorry, but you are trying to make out some conclusions, reporting that you have studied only 20 subjects. The number of the patients suffering from Tube-dysfunction is very (very...) low
Answer: We understand that the sample size is small and that we took many variables into account. However, since this is the first study to address epithelial inflammation of the epithelial remnant of the perforation and that significant results were found in the sample size used, we believe this is the beginning of a path to follow.
We add a paragraph for limitations of the study on discussion , page 9.
- Discussion: Too short for such a theme. It MUST be re-rewritten.
Answer: We've made some changes and added some information based on your feedback.
- I do not make any further comments on the References...
Answer: Thanks for your review.
Reviewer 3 Report
Comments and Suggestions for Authors
The paper is well written with appropriate references but, as pointed out, with so many variables, the sample size is small . Thus the power of this study is very limited. Also I am not sure what the clinical significance of this finding can contribute to the management or decision making in the care of these patients.
Author Response
The paper is well written with appropriate references but, as pointed out, with so many variables, the sample size is small . Thus the power of this study is very limited. Also I am not sure what the clinical significance of this finding can contribute to the management or decision making in the care of these patients.
Answer: We understand that the sample size is small and that we took many variables into account. However, since this is the first study to address epithelial inflammation of the epithelial remnant of the perforation and that significant results were found in the sample size used, we believe this is the beginning of a path to follow.
We add a paragraph for limitations of the study on discussion , page 9.
Reviewer 4 Report
Comments and Suggestions for Authors
The authors conducted a research on the outcomes after tympanoplasty in patients with tympanic membrane perforation. Epithelial inflammation plays a significant role in the outcomes of tympanoplasty, particularly in patients with chronic otitis media. After a careful study, including histopathological, the authors conclude that the presence of epithelial inflammation in the remnant of the tympanic can inhibit successful healing and lead to poorer surgical outcomes. The study is accurate and well executed, but, based on histopathological data, it would be useful to publish some images of the histological findings.
Author Response
The authors conducted a research on the outcomes after tympanoplasty in patients with tympanic membrane perforation. Epithelial inflammation plays a significant role in the outcomes of tympanoplasty, particularly in patients with chronic otitis media. After a careful study, including histopathological, the authors conclude that the presence of epithelial inflammation in the remnant of the tympanic can inhibit successful healing and lead to poorer surgical outcomes. The study is accurate and well executed, but, based on histopathological data, it would be useful to publish some images of the histological findings.
Answer: Thank you for your comment. Figure 3 has been added, with representative images of the histology. Thank you.
Round 2
Reviewer 1 Report
Comments and Suggestions for Authors
The authors attribute the relatively low closure rate (65%) to the fact that the disease of the subject patients was severe. However, there is no clear statement in the text as to how “severe cases” are defined.
Furthermore, if “epithelial inflammation,” which is the central theme of this study, is considered an indicator of severity, the fact that inflammation was confirmed in only 8 of 20 cases (40%) seems inconsistent with the claim that “the majority of cases were severe.
This point is insufficient to explain the low success rate solely in terms of the large number of severe cases.
Also, do the authors select soft tissues such as fascia as graft material in some cases?
The authors say that they determined that cartilage transplantation was optimal for the patient group in this study, which included many severe cases, but the reason for selecting severe cases as the target group is also unclear.
The specific reasons for judging them as severe cases are insufficient, and you need to clearly explain how they are related to the postoperative results.
Author Response
The authors attribute the relatively low closure rate (65%) to the fact that the disease of the subject patients was severe. However, there is no clear statement in the text as to how “severe cases” are defined.
Furthermore, if “epithelial inflammation,” which is the central theme of this study, is considered an indicator of severity, the fact that inflammation was confirmed in only 8 of 20 cases (40%) seems inconsistent with the claim that “the majority of cases were severe.
This point is insufficient to explain the low success rate solely in terms of the large number of severe cases.
Also, do the authors select soft tissues such as fascia as graft material in some cases?
The authors say that they determined that cartilage transplantation was optimal for the patient group in this study, which included many severe cases, but the reason for selecting severe cases as the target group is also unclear.
The specific reasons for judging them as severe cases are insufficient, and you need to clearly explain how they are related to the postoperative results.
Answer:
Thank you for your comments on our manuscript, in relation to your observations, we comment:
1. We clasificated the clinical severity of disease according to the index OOPS (Ossiculoplasty Outcome Parameter Staging) and MERI (Middle Ear Risk Index). The mean for the OOPS score was 2.45 (SD 1.5) and for the MERI score, it was 4 (1.8). In the group of patients with surgical failure the mean for OOPS was 3.7 (SD .95) and for MERI 5.1 (SD 1.3). that means that the most of patientes present moderate desease. They classify the disease into three stages: mild, moderate, and severe. The OOPS score ranges from 1-3, 4-6, and 7-9, and the MERI score ranges from 1-3, 4-6, and 7-12, respectively, wich means that most of patients present moderate and severe disease. (Jung), (Dash).
You can see the changes in the manuscript at:
Material and methods, page 3
Results, page 4
Discussion, pages 7-8
The bibliography that supports our answer is:
*Jung DJ, Lee HJ, Hong JS, Kim DG, Mun JY, Bae JW, et al. Prediction of hearing outcomes in chronic otitis media patients underwent tympanoplasty using ossiculoplasty outcome parameter staging or middle ear risk indices. PLoS One [Internet]. el 1 de julio de 2021 [citado el 1 de noviembre de 2023];16(7). Disponible en: /pmc/articles/PMC8321221)
*Dash M, Deshmukh P, Gaurkar SS, Sandbhor A. A Review of the Middle Ear Risk Index as a Prognostic Tool for Outcome in Middle Ear Surgery. Cureus. 2022 Nov 3;14(11):e31038. doi: 10.7759/cureus.31038. PMID: 36475203; PMCID: PMC9719032.
2. We selected cartilage graft taking on count that in our sample the mayority of patients presented tubal dysfunction (80%) and some authors refer that in this cases is better to use cartilage graft (Valiya) (Singh).
You can see the changes in the manuscript at:
Discusión, page 8
The bibliography that supports our answer is:
* Valiya V, Kumar R, Kapadia PB, Panchal AJ, Luhana MA, Tailor U. Association of Pre-operative Eustachian Tube Function with the Graft Uptake After Tympanoplasty. Indian J Otolaryngol Head Neck Surg. 2024 Feb;76(1):540-544. doi: 10.1007/s12070-023-04208-z. Epub 2023 Sep 20. PMID: 38440657; PMCID: PMC10908895.
* Singh A, Talda D, Bhutia CD, Aggarwal SK, Chakraborty P, Kumari S, Yadav S. A Prospective Randomised Comparative Study Between Cartilage and Fascia Tympanoplasty in a Tertiary Care Hospital to Look for Better Alternative in High Risk Cases. Indian J Otolaryngol Head Neck Surg. 2023 Apr;75(Suppl 1):50-59. doi: 10.1007/s12070-022-03175-1. Epub 2022 Nov 6. PMID: 37206716; PMCID: PMC10188854.
Reviewer 2 Report
Comments and Suggestions for Authors
This is a better version of the previous text, but there is still an important issue, which must be adressed. The number of the participants. Ofcourse the authors have already mentioned that as a limitation, but this can not " cure" the weakness in their methodology. I believe, that they must examine more patients and then submit the paper once again.
Author Response
This is a better version of the previous text, but there is still an important issue, which must be adressed. The number of the participants. Of course the authors have already mentioned that as a limitation, but this can not " cure" the weakness in their methodology. I believe, that they must examine more patients and then submit the paper once again.
Answer:
Thank you for your comments on our manuscript, in relation to your observations, we comment
We estimated the effect size and detected a power of 77% (Group with succes 25%vs Group without succes 57.7 %), and it tell´s us that relationships between variables or the difference between groups were high.(Cohen)
You can see the changes in the manuscript at:
Discusión: page 7.
LImitations, page 9
The bibliography that supports our answer is:
Cohen, J. (1988). Statistical power analysis for the behavioral sciences (2nd ed.). New Jersey: Lawrence Erlbaum.
Reviewer 3 Report
Comments and Suggestions for Authors
The paper confirms through statistical analysis that chronic inflammation around the epithelial edges of the perforation significantly leads to higher failure rates. The variables have been adequately addressed when drawing their conclusions.
Author Response
Thank you for your comments on our manuscript,